# Immune Activity of Polysaccharide Fractions Isolated from Korean Red Ginseng

**DOI:** 10.3390/molecules25163569

**Published:** 2020-08-06

**Authors:** Soo Hyun Youn, Sang Min Lee, Chang-Kyun Han, Gyo In, Chae-Kyu Park, Sun Hee Hyun

**Affiliations:** Laboratory of Efficacy Research, Korea Ginseng Corporation, 30, Gajeong-ro, Shinseong-dong, Yuseong-gu, Daejeon 34128, Korea; soo@kgc.co.kr (S.H.Y.); smlee@kgc.co.kr (S.M.L.); ckhan@kgc.co.kr (C.-K.H.); 20109042@kgc.co.kr (G.I.); ckpark@kgc.co.kr (C.-K.P.)

**Keywords:** Korean red ginseng, red ginseng acidic polysaccharide (RGAP), immune activity, antibody-forming cells (AFCs), macrophage

## Abstract

Korean red ginseng (KRG)’s pharmacological efficacy and popular immunomodulatory effects have already been proven in many studies; however, the component of KRG that is effective in immune activity has not been studied before. Therefore, this study extracted and separated KRG for an immune activity comparison. In the water fraction obtained by extracting KRG powder with water, a red ginseng neutral polysaccharide (RGNP) fraction and a red ginseng acidic polysaccharide (RGAP) fraction were obtained. Each fraction was orally administered for 10 days to mice with reduced immunity, and the number of IgM antibody-forming cells (AFCs) in splenocytes was measured to compare the immune activity of the water fractions. The results showed that the RGAP fraction has the greatest number of AFCs. To set the optimal dose of the RGAP fraction, which had the highest immune activity, the AFCs, macrophage activity, and splenocyte subtype in the mice were analyzed. As a result, the number of AFCs was significantly increased in the RGAP fraction compared to RGNP. The intraperitoneal macrophage phagocytosis activity and the number of T cells, B cells, and macrophages in the spleen increased significantly. It can, therefore, be confirmed that immune activity increases by a fraction containing higher RGAP content, and we hypothesize that RGAP activates immune activity.

## 1. Introduction

*Panax ginseng* C.A. Meyer (P. ginseng), which belongs to the genus *Panax* (ginseng), has been traditionally used in Asia, including Korea, China, and Japan, for thousands of years, not only in nutritional tonics, but also in remedies for different diseases such as autoimmune diseases, diabetes, and cancer. The effects of ginseng on cells, animals, and humans have been scientifically proven by many studies, with its efficacy largely described in terms of its adaptogen activity, intending to maintain metabolic homeostasis by normalizing the overall vital function through higher nonspecific resistance to external stress [1,2,3]. 

P. ginseng is referred to in different ways depending on its processing method. Freshly harvested P. ginseng is called fresh ginseng, dried ginseng is called white ginseng, and steamed and dried ginseng is called red ginseng. KRG has been recognized by the Ministry of Food and Drug Safety, formerly known as the Korea Food and Drug Administration (KFDA), as a health functional food with six health benefits: boosting immunity, overcoming fatigue, improving memory, improving blood circulation, alleviating women’s menopausal symptoms, and promoting antioxidant activity [4]. Numerous studies on the pharmacological efficacy of KRG have been published through in vitro, in vivo, and clinical trials. In a study by Kim et al. [5], patients with colorectal cancer who received mFOLFOX-6 chemotherapy were randomly administered either KRG 2000 mg/day or placebo for 16 weeks. Compared to the placebo group, the KRG group had reduced cancer-related fatigue [5]. Moreover, 12-week KRG administration to menopausal women significantly increased superoxide dismutase (SOD) activity [6]. Furthermore, healthy subjects who received KRG had improved SOD, glutathione peroxidase, and catalase [7]. Prehypertensive subjects who were administered with KRG showed more controlled systolic blood pressure (BP), diastolic BP, and plasma lipoprotein-associated phospholipase A2 activity compared with placebo [8]. In addition, in previous studies, KRG is shown to affect blood sugar, lessen menopausal symptoms, and slow the depletion of CD4 T cells in human immunodeficiency virus type 1 infected patients [9,10,11].

As a result of the partial structural change caused by the detachment of a glycosyl residue at the C-20 position of ginsenoside and the isomerization of a hydroxyl group by thermal decomposition during KRG processing, 14 saponin components, including ginsenoside Rg5, Rh2, Rh4, and Rs1, as well as maltol and arginine—fructose—glucose (Arg–Fru–Glc) with antioxidant effects, which are only found in KRG, are largely produced [1]. In addition to saponins, fat-soluble components like polyacetylenes, alkaloids, phenolic compounds, lignans, and sesquiterpenes, and water-soluble components like carbohydrates, nitrogen-containing compounds, minerals, and vitamins are found in KRG. Of the carbohydrates, acidic polysaccharides are polysaccharides with a molecular weight of 12,000 or more combined with galacturonic acid, glucuronic acid, and mannuronic acid, and they are the most studied nonsaponin compounds [12]. Compared to white ginseng (0.6–0.8%), KRG (4.7–7.4%) showed a much higher component of acidic polysaccharides, which means that they can be produced in large amounts in the process of manufacturing KRG [13]. Some of the known pharmacological activities include the alleviation of hypertension and hyperlipidemia, prevention of blood clotting, hematopoietic function, and a reduction in nutritional and metabolic destruction in alcohol addiction, leading to the improvement of the liver’s condition [14,15,16,17,18]. In addition, acidic polysaccharides prevent acetaminophen-induced liver toxicity, inhibit lipolysis of cancer toxin hormones, alleviate symptoms of digestive organ cancer, carry anticancer effects by their activation of macrophages and natural killer cells, reduce the cell adherence of pathogens like *Brucella abortus*, and protect against radiation [19,20,21,22,23,24,25,26,27]. 

Today, people are exposed to an environment in which immune functions can easily deteriorate because of environmental pollutants, medicinal side effects, diseases, stress, and irregular or Westernized diets. It has been reported that the KRG acidic polysaccharide (RGAP) is a great functional material with immune—physiological activity that reduces the change in immune function or restores its function [28,29]. Immune response is a protective action that eliminates the disease-prone environment induced by the invasion of pathogens or foreign objects, and repairs the body. Such defense mechanisms are divided into innate and adaptive immune responses. This innate immune response is the first barrier of defense against various infections caused by external organisms or antigenic substances, and it includes physical/chemical defenses like the acidity of the skin, mucous membranes, and stomach, the activity of immune cells like dendritic cells, as well as Langerhans cells, macrophages, and natural killer cells (NK cells), and the activation of blood proteins such as complements and inflammatory mediators. This response involves signaling the presence of infections and stimulating the adaptive immune response, which is immunity acquired from exposure to an antigen, and carries the following characteristics: specificity, diversity, memory, and self-recognition/non–self-recognition. In addition, the adaptive immune response is classified into cell-mediated and humoral immune response according to the antigen removal reaction. Cellular immunity is an immune response caused by intercellular interactions through the mediation of helper T cells (CD4+), cytotoxic T cells (CD8+), B cells, and antigen-presenting cells. Humoral immune responses consist of reactions with B cell-derived antibodies produced specific to the structure of the antigen, and responses mediated by complements in serum that are synthesized in the liver and macrophage and then separated [30,31]. However, although various studies on the immune activity of KRG have been done, the components that contribute to the enhancement of immunity have not been identified. It can be inferred from the existing literature that RGAPs carry out immune-physiological activity, but no studies have systematically proven such a hypothesis [3]. Therefore, to assess the functional components of six-year-old KRG, the immune-enhancing effects of a fraction made using ethanol and distilled water (DW) in our previous study were compared [32]. 

A summary of the previous study results is as follows. The homogenized six-year-old KRG was extracted with 100% ethanol to obtain an ethanol fraction and residue (R1). When the immune enhancement effects of the two fractions were compared, R1 was better in function, and R1 was reextracted with DW to obtain a water fraction (W) and secondary residue (R2). It was then confirmed that the immune-enhancing effect of W is better than that of R2, and as W had more RGAPs than other fractions, it can be presumed that the immuno-enhancing activity of W is due to RGAPs. Therefore, the present follow-up study extracted more pure polysaccharides using a column from W, which had the best immune-enhancing effect, and compared the immune activity of polysaccharides.

## 2. Results and Discussion 

### 2.1. Component Analysis

Table 1 shows the composition of W, RGNP, and RGAP. The content of glucose, fructose, sucrose, maltose, acidic polysaccharides, and ginsenosides (total percentage of Rg1, Re, Rf, Rg2(S), Rg2(R), Rb1, Rc, Rb2, Rd, Rg3(S), and Rg3(R)) in W was 2.4%, 1.7%, 1.53%, 5.1%, 7.46%, and 6.11%, respectively). The content of glucose, fructose, sucrose, and maltose in RGNP was 1.14%, 1.70%, 1.70%, and 5.80%, respectively, and acidic polysaccharides were not detected. Meanwhile, the content of ginsenoside was 5.51% in RGNP, which was slightly lower compared to its content in W. The analysis of RGAP showed the detection of acidic polysaccharides (57.25%) with no other components. 

### 2.2. Weight Change of Immunity Organs 

W, RGNP, and RGAP were orally administered for 10 days, and cyclophosphamide (CY) was administered 5 days before the autopsy, followed by weight measurements of immunity organs. In all the groups administered with CY, the relative and absolute weight of immunity organs were significantly reduced (*p* < 0.05). Absolute organ weight showed a 25.2% reduction in the weight of the spleen in the CY administration group compared to the normal group (*p* < 0.01), and the administration of W, RGNP, and RGAP induced weight recovery to 4.6%, 9.7%, and 11.7%, respectively, compared to the CY control, but there was no significant change. The absolute weight of the thymus decreased by 23.6% compared to the normal group (*p* < 0.01), and the administration of W, RGNP, and RGAP induced recovery rates of 0.9%, 4%, and 7.3%, respectively, compared to the CY control, but there was no significant change. The relative weight of the spleen decreased by 21.3%, and the thymus by 26.1% by administration of CY compared to the normal group (*p* < 0.01). Administration of RGNP and RGAP induced an increasing tendency in the relative weight of the spleen by 2.7% and 8.1%, respectively. The relative weight of the thymus showed recovery rates of 5.9%, 11.8%, and 11.8% by administration of W, RGNP, and RGAP, respectively, but there was no significant change (Table 2).

### 2.3. IgM Antibody-Forming Cells (AFCs) in Splenocyte against SRBCs 

When the splenocytes of mice immunized by SRBCs are cultured in mixture with SRBCs, antibody-producing B cells release immunoglobulins (Ig). The released Ig binds to the red blood cells, and the complements act to lyse the red blood cells around the antibody-producing cells and form plaques. To compare the ability of W, RGNP, and RGAP to form AFCs in animals with immunity reduced by CY, each sample was orally administered for 10 days. Moreover, five days before the autopsy, 50 mg/kg of CY was intraperitoneally injected to all animals except the normal group. On the day after CY administration, SRBCs were intraperitoneally administered to all animals to induce an immune response, and the AFC formation in the spleen was compared three days after autopsy. The total number of splenocytes was significantly reduced to 41.9%, 43.8%, 48.6%, and 39% by administration of CY control, W, RGNP, and RGAP, respectively, compared to the normal group (*p* < 0.01). However, there was no intergroup difference by sample administration (Figure 1A). The number of AFCs per 1 million splenocytes was significantly decreased to 68% in the CY control compared to the normal group (*p* < 0.01) and was increased by 63.2%, 42.2%, and 75.1% in the W, RGNP, and RGAP groups compared to the CY control. In particular, W and RGAP showed significant increases (*p* < 0.05, Figure 1B). AFCs per spleen showed a decrease of 81.8% in the CY control compared to the normal group (*p* < 0.01), and an increase of 59.8%, 30.4%, and 81.5% by administration of W, RGNP, and RGAP compared to the CY control, of which RGAP showed a significant increase (*p* < 0.01, Figure 1C). RGAP, which showed the highest AFC formation, had an acidic polysaccharide content of 57.25%, which was 7.7 times higher than that of W. After RGAP was administered orally at doses of 50, 100, and 200 mg/kg for 10 days, AFC formation was measured. The total count of splenocytes in CY control was significantly decreased to 42.2% compared to the normal group (*p* < 0.01), and there was no significant change by dose-specific administration of RGAP compared to the CY control (Figure 2A). The number of AFCs per one million splenocytes was significantly decreased to 72.8% in the CY control compared to the normal group (*p* < 0.01) and was increased by 3.4% and 60.7%, and 93.7%, in a dose-dependent manner in the RGAP 50, 100, and 200 mg/kg administration groups compared to the CY control. In particular, administration 100 and 200 mg/kg of RGAP showed a significant increase (*p* < 0.01, Figure 2B). AFC activation per spleen showed a reduction rate of 82.9% in the CY control compared to the normal group (*p* < 0.01). There was no change in the 50 mg/kg RGAP administration group compared to the CY control, and the AFC activity was significantly increased to 57.3% and 129.3% by the administration of 100 and 200 mg/kg of RGAP (*p* < 0.01, Figure 2C).

### 2.4. Phagocytosis Activity of Peritoneal Macrophage 

The macrophage secretes the cytokine, which is a signaling protein and is responsible for intracellular pathogens, autoimmunity, and inflammatory effects through phagocytosis and antigen production. In experimental animals with immunity reduced by CY, the effect of dose-specific RGAP administration on macrophage phagocytosis was measured. The phagocytosis of macrophage decreased to 25.9% in the CY control compared to the normal group (*p* < 0.01) and was significantly increased by 17.5%, 23.1%, and 25% in a dose-dependent manner by the administration of 50, 100, and 200 mg/kg of RGAP (*p* < 0.05; Figure 3).

### 2.5. Distribution of Spleen Lymphocyte 

When cells in charge of innate immunity present an antigen on the surface even after removing foreign antigens, the cytotoxicity T cells recognize the antigen and remove it while also destroying the infected cell. As such, after recognizing the presented antigen, the helper T cells differentiate into different cell types like Th1 and Th2 and decide whether to suppress or continue with the immune response. This leads to signaling the immune cells like the macrophage and B cell, playing an immune cell support role. B cells produce antibodies immediately if they have information about the antigens presented by helper T cells and differentiate into memory cells if they do not have the information. The effect of RGAP on splenocytes in test animals with immunity reduced by CY was measured by fluorescence-activated cell sorting (FACS) analysis. The number of CD3+ T cells was significantly decreased to 58.8% in the CY control compared to the normal group (*p* < 0.01). Compared to the CY control, the number increased by 50, 100, and 200 mg/kg of RGAP administration by 19.0%, 47.6%, and 63.5%, respectively, and, in particular, the 100 and 200 mg/kg administration groups showed significant increases (*p* < 0.01; Figure 4A). In addition, the number of B cells decreased by 87.4% in the CY control compared to the normal group (*p* < 0.01) and significantly increased by 30.8%, 92.3%, and 123.1% by 50, 100, and 200 mg/kg RGAP administration compared to the CY control (*p* < 0.01; Figure 4B). The number of macrophages also decreased by 90.2% in the CY control compared to the normal group (*p* < 0.01). Compared to the CY control, 50, 100, and 200 mg/kg RGAP administration increased by 18.2%, 72.7%, and 118.2%, respectively. In particular, 100 and 200 mg/kg administration groups showed a significant increase (Figure 4C). The administration of 100 and 200 mg/kg RGAP restored the CY-induced reduction in T cell, B cell, and macrophage count.

### 2.6. Discussion

KRG is an exemplary herbal medicine known for its pharmacological effects in Asia, especially Korea and China [33], as many studies have shown its role as herbal medicine in controlling anticancer, cardiovascular disease, immunity, diabetes, and skin activity [34,35,36,37]. In particular, its most representative pharmacological effect is the regulation of the immune system and protection from external infections (viruses, cells, diseases) [29]. Active ingredients of KRG include ginsenosides, flavonoids, polyphenols, and polysaccharides, which are generally divided into RGNP and RGAP [38]. The acidic polysaccharide refers to a polysaccharide with a molecular weight of 10,000 to 15,000 containing a large amount of acidic sugar, like galacturonic acid, glucuronic acid, and mannuronic acid. Such RGAPs are also known to have a greater effect on the immune system than RGNPs [26].

A previous study has found that the acidic polysaccharides isolated from Panax ginseng promote the proliferation of T cells and B cells and affect the immune system [22]. In addition, cytotoxic activity against B16 melanoma cells was induced when macrophages were treated with polysaccharides, and the secretion of tumor necrosis factor-α (TNF-α), interleukin-1β (IL-1β), IL-6, and interferon-γ (IFN-γ) was promoted [39]. Moreover, many studies confirmed that RGAPs have various immune activity effects [40,41]. Although there are many studies on the immune activity of acidic polysaccharides, these studies consider the crude form of the polysaccharide. In addition, no systematic study has been done on the preparation of fractions and comparison of the activity of each fraction to identify components that show immune activity in a single root of KRG, as in this study.

This study is meaningful in that it systematically identified the components contributing to the immune activity of KRG. Fractions were prepared from KRG powder for each step, and the immunological activities of the fractions were compared to select a superior fraction.

In this study, W, RGNP, and RGAP did not significantly affect the restoration of thymic and splenic weights, which were reduced by deteriorated immunity. However, note that they showed a tendency to recover weight compared to the CY control. The recovery rate of the spleen and thymus was lower in W, which had a higher acidic polysaccharide content than RGNP. This was attributed to individual differences, and there was no significant difference between the two groups. Hyun et al. [32] also reported that KRG contributed to the recovery of thymic and splenic weights reduced by the weakened immune system [42]. RGAP turned out to have the highest AFC formation, followed by W and then RGNP. This meant that a fraction with more acidic polysaccharides is more capable of producing antibodies. As shown in this study, Park et al. [28] also demonstrated that acidic polysaccharides isolated from KRG promoted antibody production in mice. As a result, it was found that the higher content of acidic polysaccharides showed relatively higher AFCs when compared with each fraction’s components. This finding could lead to the reasoning that among different KRG components, RGAPs are polysaccharides that contribute to immunoreactivity.

This study reviewed further work regarding RGAP’s immune activities such as AFCs, phagocytosis activity, and splenocyte immune cell subtype analysis. Here, RGAP groups showed significantly increased AFCs in all groups dose dependently compared to the immunosuppressed CY control. Lee et al. [43] studied KRG polysaccharides’ immune effects, and their results showed increased AFCs compared to the negative control. The present study shows that phagocytosis activity significantly increased compared to the CY control in all doses of RGAP groups. Phagocytosis is a crucial physiological process that is characterized by the ingestion of foreign particles and the killing of bacteria by phagocytic leukocytes, including macrophages; it serves as the first line of host defense against pathogens. This study’s results indicated that RGAP potentially enhanced phagocytosis to defend the host from infection. Moreover, the splenic absolute number of CD3+ T cells, CD45R/B220+ B cells, and CD11b+ macrophage cells was significantly increased at RGAP 100, 200 mg/kg groups. These results indicate that RGAP exhibits immune-enhancing effects. Byeon et al. [33] investigated the mechanism by which RGAP stimulates the immune response. RGAP treatment on macrophages showed that the former activates transcription factors, such as the nuclear factor kappa-light-chain-enhancer of activated B cells (NF-κB), and activator protein 1 (AP-1) and upstream signaling enzymes, such as extracellular signal-related kinase (ERK), to activate macrophages and induce immune activity. Similarly, in this study, RGAP was found to activate immune cells (T cells, B cells, and macrophages) and increase their number to induce immune activity. 

The clear composition and structural activity of acidic polysaccharides have not yet been identified. Moreover, acidic polysaccharides’ immune active mechanisms are still insufficient. Therefore, further study needs to be done on the components of polysaccharides derived from KRG. The RGAP was found to have excellent immune activity in this study, showing the possibility of development into a functional food and pharmaceutical supplement, replacing the conventional immunomodulatory substances that present toxicity and side effects.

## 3. Methods

### 3.1. Manufacture of Korean Red Ginseng

In this experiment, Panax ginseng was grown in Gyeonggi-do, Icheon (South Korea), for six years. The manufacturing process of Korean red ginseng was certified in April 2017 by the International Organization for Standardization (ISO) and officially approved globally (ISO 19610); the process is as follows: sorting six-year-old fresh ginseng roots based on the thickness of the main root before washing; steaming at a temperature of 90 °C to 100 °C for at least 80 to 100 min for starch gelatinization; and, lastly, drying with hot air at 45 °C to 55 °C and in the sun until the moisture content is 15.5% or less. KRG powder is a product of pulverizing dried KRG (120 mesh or less) with a moisture content of less than 3% to 6%. 

### 3.2. Production of Fractions

To manufacture fractions, 1 kg of KRG powder and 3 L of 100% ethanol (EtOH) were mixed and extracted using ultrasonic extraction 2 h at 40 °C. The supernatant was collected after extraction. Precipitated residue was added to 2 L of 100% EtOH and extracted using ultrasonic waves for 2 h at 40 °C again. The above method was performed three times. Upper solutions were collected, vacuum evaporated 50 °C, and lyophilized for three days to make the ethanol fraction (E). Residues were collected and dried with an air dryer at 45 °C for 12 h, from which Residue 1 (R1) was obtained. R1 was dissolved with DW and extracted for 4 h using ultrasonic waves at room temperature. Upper solutions were obtained after centrifugation, and the same procedure was performed two more times. Afterward, upper solutions were collected, vacuum evaporated in a 60 °C water bath, and lyophilized to obtain the water fraction (W). In W separated from KRG, KRG neutral polysaccharides (RGNPs) and RGAPs were fractionated according to the extraction method used by Zhang et al. [44] as follows. A glass column (4 × 40 cm) was filled with 300 g of diethylaminoethyl cellulose (Sigma Aldrich Co., St. Louis, MO, USA), which was thoroughly washed with triple DW to remove air bubbles and stabilized. W was poured into the filled glass column, and 5 L of triple DW was added to elute the neutral fraction at 50 mL/min. The eluted solution was lyophilized to obtain the RGNP fraction. To the column with the neutral fraction removed, 3 L of 1.0 M NaCl was added to elute the acidic fraction at 50 mL/min, and dialysis was performed to remove the salt. The semipermeable membrane used was the Spectra/Por Dialysis Membrane (molecular weight cut-off: 3500; Spectrum Laboratories, Inc., Rancho Dominguez, CA, USA). An acidic fraction eluate was added to the semipermeable membrane, and the triple DW was changed once a day, followed by a three-day dialysis to remove salt and low molecular substances. Then, it was lyophilized to produce an RGAP fraction (Figure 5). Concerning the yields of each fraction, they were 84.6% for R1, 45.3% for W, 37.5% for RGNP, and 5.5% for RGAP. As each fraction was prepared, a loss of fraction samples occurred, and its rate reached 2% to 6%.

### 3.3. Component Analysis 

The analysis of free sugar was performed according to Joo et al. [45] with some modifications. Analysis of four types of free sugar—glucose, fructose, sucrose, and maltose—was carried out by extracting 0.1 g of the sample with 10 mL of DW, taking 1 mL of the extracted sample and making the total volume to 10 mL. The sample was centrifuged for 10 min at a speed of 1800 rcf/min and filtered through a 0.2 μm syringe filter for analysis. The high-performance ion chromatography used for the analysis was connected to the pulsed amperometric detector (PAD), Au working electrode, and Ag/AgCl reference electrode using the ICS300 system, and the analysis column used was CarboPac PA-1 (250 × 4 mm; Dionex, Sunnyvale, CA, USA). For the mobile phase, 250 mM NaOH and H_2_O were used, and the solvent ratio was sequentially changed from 93% to 50% (35 min), 0% (45 min), and 93% (60 min). The column temperature was maintained at 30 °C to minimize component change. The mobile phase flow rate was set to 1.0 mL/min, and the sample injection amount was maintained at 10.0 μL/min for analysis. 

For the analysis of 11 kinds of ginsenosides, samples were weighed in a centrifugal tube (BioLogix Group, Jinan, Shandong, China) and shaken vigorously after the addition of 10 mL of 70% MeOH. Extraction was performed in an ultrasonic cleaner (60 Hz, Wiseclean; Daihan Scientific, Seoul, Korea) for 30 min. After ultrasonic extraction, centrifugal separation (Legend Mach 1.6R; Thermo, Frankfurt, Germany) was performed for 10 min at 1800 rcf. The resulting supernatant solution was filtered (0.2 mm, Acrodisc; Gelman Sciences, Ann Arbor, MI, USA) and injected into the UPLC- photodiode array detector (PDA) system (Waters Co., Milford, MA, USA). The instrumental analysis was performed by a Waters ACQUITY UPLC system (Waters, Milford, MA, USA) composed of a binary solvent manager, sample manager, and photodiode array detector (PDA). The chromatographic separation was accomplished on an ACQUITY BEH C18 column (2.1 mm × 50 mm, 1.7 um; Waters). The column temperature was 40 °C. The binary gradient elution system consisted of deionized water (A) and acetonitrile (B). The UPLC gradient conditions were as follows: 0.5–14.5 min (15–30% B), 14.5–15.5 min (30–32% B), 15.5–16.5 min (32–40% B), 16.5–17.0 min (40–55% B), 17.0–21.0 min (55–90% B), 21–25 min (90–15% B), and 25–27 min (15% B). The flow rate was set at 0.6 mL/min, and the sample injection volume was 2.0 μL. The 11 ginsenosides were detected by PDA at 203 nm [46]. The content of acidic polysaccharides was measured by the carbazole method using glucuronic acid as a standard substance [47]. First, 50 mL of DW was added to a beaker containing the sample before shaking it to obtain a homogeneous solution. Next, 9 mL of ethanol was added to 2 mL of the solution, which is then left at 4 °C for 4 h. After centrifugal separation, the resulting precipitate was dissolved in 2 mL of DW. The centrifugation was carried at 300 r/min for 10 min at 4 °C. Afterwards, the supernatant was discarded, and the precipitated residue was dissolved in 2 mL of DW. The sample should be diluted using the calibration curve. Then, 0.5 mL of the diluted sample solution was taken before adding 0.25 mL of the carbazole reagent and 3 mL of sulfuric acid thereto for colorimetric analysis. The developed sample solution was incubated for 5 min in a water bath operating at 80 °C, and then cooled in cold water for 15 min. Next, 200 μL of the sample solution each was injected into a well plate with 96 wells to measure the optical density values at 525 nm. The composition of W, RGNP, and RGAP is shown in Table 1.

### 3.4. Reagents and Analysis Devices 

Sheep red blood cells (SRBCs) were purchased from South Pacific Sera Co. (Timaru, New Zealand), and Earle’s balanced salt solution (EBSS), DEAE-dextran, agar, 2-mercaptoethanol, guinea pig complement, agar, and cyclophosphamide (CY) reagents were purchased from Sigma-Aldrich Co. (St. Louis, MO, USA). Roswell park memorial institute medium (RPMI1640), fetal bovine serum (FBS), penicillin–streptomycin, L-glutamine, and hydroxyethyl piperazine ethane sulfonic acid (HEPES) buffer were purchased from Gibco Co. (Rockville, MD, USA), and ammonium-chloride-potassium (ACK) lysis buffer was purchased from Lonza Co. (Walkersville, MD, USA). Antibodies, including purified anti-mouse CD16/CD32 Fc receptor, peridinin chlorophyll-a protein (PerCP)-conjugated anti-mouse CD3e (clone: 145-2C11), R-phycoerythrin (R-PE)-conjugated anti-mouse CD45R/B220 (clone: RA3-6B2), and fluorescein isothiocyanate (FITC)-conjugated anti-mouse CD11b (clone: M1/70) were purchased from BD Pharmingen Inc. (San Diego, CA, USA), and the phagocytosis assay (IgG FITC) kit was purchased from Cayman Chemical Co. (Ann Arbor, MI, USA). 

### 3.5. Animals 

For this study, six-week-old male Balb/c mice were purchased from Koatech (Pyeongtaek, Korea) and were stabilized for seven days. During the stabilization period, general symptoms were monitored, and normal animals with an average weight of 22 to 23 g were selected. Eight mice per group were used in the experiment. The temperature of the breeding room was maintained at 20 ± 2 °C, relative humidity at 50 ± 5%, the number of ventilation at 10 to 12 times/h, the lighting time from 7:00 a.m. to 7:00 p.m., and the illuminance was between 150 and 300 Lux. Water and food (Feedlab, Guri, Korea) were provided ad libitum. This study was carried out following the ethical regulations of the KT&G Institutional Animal Care and Use Committee (KT&G 10-015, KT&G 11-002, KT&G 11-005).

### 3.6. Sample Administration 

To compare the immune activity of the W, RGNP, and RGA fractions, 40 mice were divided into 5 groups: normal, CY control, W, RGNP, and RGAP. Then, 100 mg/kg of each sample was orally administered daily for 10 days. 

For the antibody-forming cell activity test according to the dose of RGAP, 40 mice were divided into 5 groups: normal, CY control, 50 mg/kg of the RGAP fraction, 100 mg/kg of the RGAP fraction, and 200 mg/kg of the RGAP fraction. The mice were orally administered the test substance daily for 10 days. 

For macrophage activity and splenocyte subtype analysis, 40 mice were divided into 5 groups: normal, CY control, 50 mg/kg of the RGAP fraction, 100 mg/kg of the RGAP fraction, and 200 mg/kg of the RGAP fraction, following a randomized block design. Test substances were orally administered daily.

### 3.7. Weight Measurement of Immune System Organs

On the day of the autopsy, the body weight of the animals was measured. Then, animals were anesthetized with CO_2_ gas, and venesection was carried out, followed by extraction of the liver, spleen, and thymus to measure the weight. 

### 3.8. Antibody-Forming Cells (AFCs) in Splenocytes against SRBCs 

Sheep blood was refrigerated and used within one week to measure the number of IgM plaque-forming cells or AFCs. While orally administering each sample, all groups except the normal group were intraperitoneally administered 50 mg/kg of CY five days prior to the autopsy, and the normal group was administered saline. On the next day, SRBCs underwent centrifugal washing three times with EBSS (290 rcf, 10 min, 4 °C), and the concentration was adjusted with EBSS to have 5 × 10^8^ cells/mL (about 5%) of SRBCs. Afterwards, 0.5 mL of the solution was intraperitoneally injected in all animals to induce an immune response. The spleen was extracted and stored in 5 mL of ice-cold EBSS medium. The prepared spleen was lightly crushed with a sterile syringe and passed through a 40 μm nylon mesh, followed by centrifugation (290 rcf, 10 min, 4 °C) to remove the supernatant. After the addition of 3 mL of EBSS medium and suspension, 100 μL was taken and placed in 2.9 mL of EBSS medium of a new tube to make a 30-fold diluted cell suspension solution. The less-than-one-week refrigerated SRBCs were taken out immediately before use, and the supernatant was removed after three instances of centrifugal washing (290 rcf, 10 min, 4 °C) with EBSS medium. Agar was put into EBSS medium (with 15 mM HEPES) to reach 0.5% and was boiled. Then, 3% DEAE-dextran was added to completely dissolve it and prepare the agar medium. The medium was maintained in a 48 °C water bath. In a round-bottom tube (12 × 75 mm), 350 μL of the agar medium, 100 μL of splenocyte cell suspension, 25 μL of SRBCs, and 25 μL of guinea pig complement were added and mixed. Then, 200 μL was dropped into a petri dish (100 mm diameter) and covered with a cover glass (24 × 40 mm). After about 20 min, when the agar is solidified, it was incubated for 4 h in an incubator (Vision Scientific Co., Daejeon, Korea) at 37 °C and 5% CO_2_ to induce plaque formation. The number of AFCs was calculated by counting the number of plaques using an electric stereomicroscope (Stereo Discovery V8, Carl Zeiss AG, Oberkochen, Germany), and the number of cells in splenocyte suspension was counted using a microscope (IX81-DSU system, Olympus, Tokyo, Japan). The cell count in the splenic cell solution was then estimated, after which the result was converted to the number of AFCs per 1 × 10^6^ splenocytes or AFCs per total splenocytes.

### 3.9. Phagocytosis Activity of Intraperitoneal Macrophage

While administering RGAP fractions orally by dose, the immunity of all groups except the normal group was reduced by administering 50 mg/kg of CY intraperitoneally three days before the autopsy. The normal group was administered with saline. On the day of the autopsy, animals were anesthetized with CO_2_ gas, and macrophages were collected after injecting 5 mL of cold PBS into the abdominal cavity. The collected cells were dispensed into a culture petri dish (120 mm diameter) and cultured in a 5% CO_2_ incubator (Vision Scientific Co., Daejeon, Korea) at 37 °C. After 2 h, cells that were not attached to the plate were removed by washing, and the attached macrophages were isolated for use. Afterward, 1 × 10^6^ cells/mL of the isolated macrophages were suspended in RPMI 1640 complement medium and cultured for 24 h. The phagocytosis activity of macrophage was measured using the phagocytosis assay kit (Cayman, 500290) under the conditions of 485 nm excitation and 535 nm emission.

### 3.10. Distribution of Spleen Lymphocyte 

Spleen was obtained from the mice after intraperitoneal macrophage collection and placed in 5 mL of RPMI1640 medium and stored on ice. The prepared spleen was made into a single-cell suspension using a 40 μm nylon mesh. The supernatant was removed by centrifugation (290 rcf, 5 min, 4 °C), and 2 mL ACK lysis buffer was added, followed by gentle tapping for 1 min for elution. Then, 5 mL of RPMI1640 medium was added, and centrifugally washed twice (1200 rpm, 5 min, 4 °C). After the suspension of the splenocytes in RPMI1640 complete (10% FBS, 2 mM L-glutamine, 100 unit/mL penicillin, 100 μg streptomycin, 5 × 10^−2^ mM 2-mercaptoethanol, HEPES buffer) medium, the number of cells was measured. The cell number was adjusted to 1 × 10^6^ cells/tube. Afterwards, cells were incubated with the CD16/CD32 Fc receptor (1 μg/tube) for 15 min at 4 °C to prevent nonspecific binding. To count the number of T cells, 1 μg/tube of PerCP-conjugated anti-mouse CD3e was followed by 30 min staining on ice. For B cell and macrophage analysis, R-PE-conjugated anti-mouse CD45R/B220 and FITC-conjugated anti-mouse CD11b were added and stained on ice. Then, the samples were washed with 1 mL of PBS three times, and 0.3 mL of PBS was added to each tube and thoroughly mixed. Flow cytometry (BD FACSCantoⅡ, BD Biosciences Co., Franklin Lakes, NJ, USA) was used for the analysis. 

### 3.11. Statistical Analysis

The results of each test were expressed as mean ± standard deviation, and when significance was observed by a one-way ANOVA test, Dunnett’s multiple comparisons were performed as a post-hoc analysis. SPSS Program (Windows Version 15.0, SPSS Inc., Chicago, IL, USA) was used for all statistical analyses, and a *p* value of 0.05 or less was determined as statistically significant.

## 4. Conclusions

KRG was fractionated with ethanol and water, the component content of each fraction was analyzed, and the immune activity of each fraction in the animal was compared. As a result, the more acidic the polysaccharide content was, the higher the immune activity was. It can be assumed that the immunoactive component of KRG is an acidic polysaccharide. In addition, this result is thought to support the results of many papers that have suggested that the immunoactive component of KRG is, indeed, an acidic polysaccharide.

## Figures and Tables

**Figure 1 molecules-25-03569-f001:**
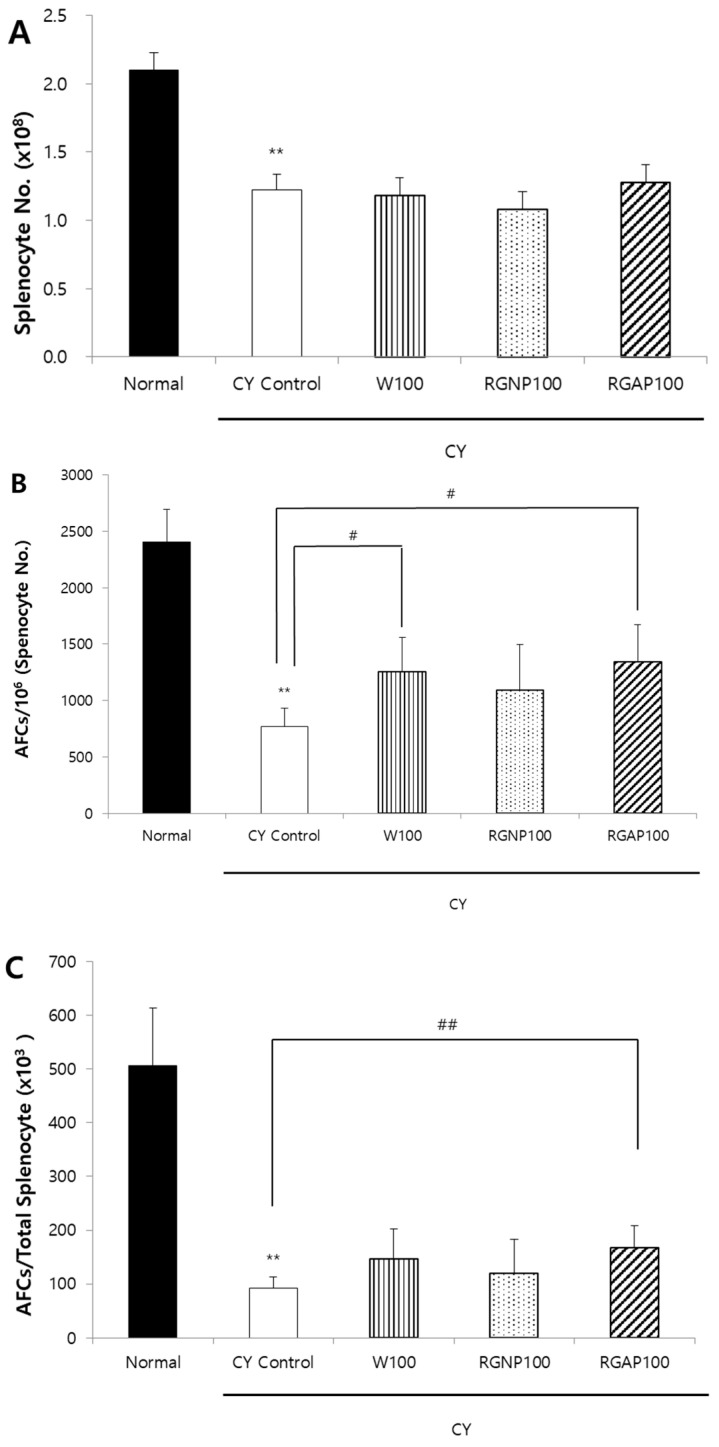
Antibody forming cells to SRBCs and splenic populations compared to Korean red ginseng powder fractions in male Balb/c mice. (**A**) Absolute splenocyte number; (**B**) AFCs per 10^6^ of splenocyte; and (**C**) AFCs per total splenocyte. Note: Mice orally administered with 100 mg/kg of W, RGNP, and RGAP for 10 days, respectively. Normal and CY control groups were administered with distilled water. Except for normal group, CY was administered intraperitoneally as a single dose of 50 mg/kg five days before sacrifice. On the following day of CY injection, mice were immunized with 2.5 × 10^8^ SRBCs per mouse in 0.5 mL EBSS by intraperitoneal injection. The results are presented as the mean ± SD of eight animals per group. Values are significantly different from normal at ** *p* < 0.01. Significant differences from CY control groups presented by ^#^
*p* < 0.05 or ^##^
*p* < 0.01. CY: cyclophosphamide, EBSS: Earle’s balanced salt solution, W: water fraction, RGNP: red ginseng neutral polysaccharide, RGAP: red ginseng acidic polysaccharide, SRBC: sheep red blood cell.

**Figure 2 molecules-25-03569-f002:**
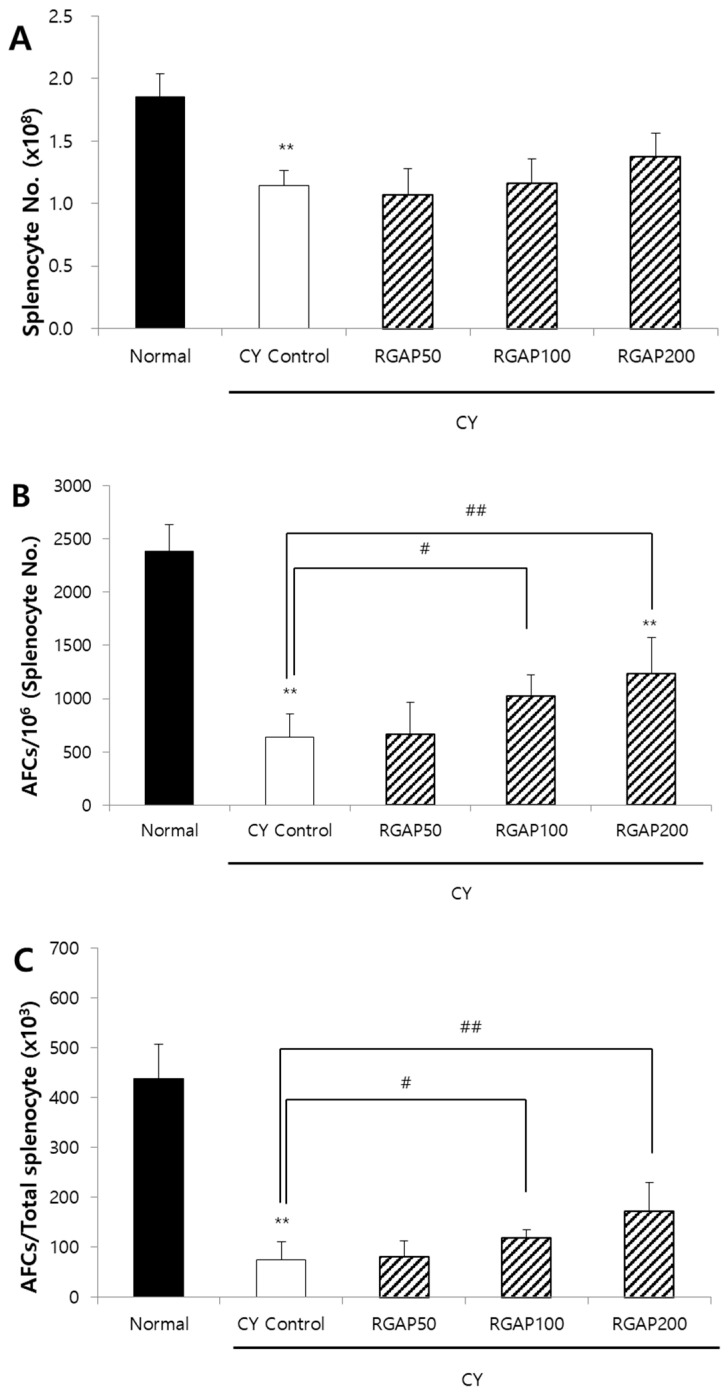
Antibody forming cells to SRBCs and splenic populations in male Balb/c mice. (**A**) Absolute splenocyte number; (**B**) AFCs per 10^6^ of splenocyte; and (**C**) AFCs per total splenocyte. Note: Mice orally administered with 50, 100, 200 mg/kg of RGAP for 10 days, respectively. Normal and CY control groups were administered with distilled water. Except for normal group, CY was administered intraperitoneally as a single dose of 50 mg/kg five days before sacrifice. On the following day of CY injection, mice were immunized with 2.5 × 10^8^ SRBCs per mouse in 0.5 mL EBSS by intraperitoneal injection. The results are presented as the mean ± SD of eight animals per group. Values are significantly different from normal at ** *p* < 0.01 and significantly different from CY control groups presented by ^#^
*p* < 0.05 or ^##^
*p* < 0.01. CY: cyclophosphamide, EBSS: Earle’s balanced salt solution, RGAP: red ginseng acidic polysaccharide, SRBC: sheep red blood cell.

**Figure 3 molecules-25-03569-f003:**
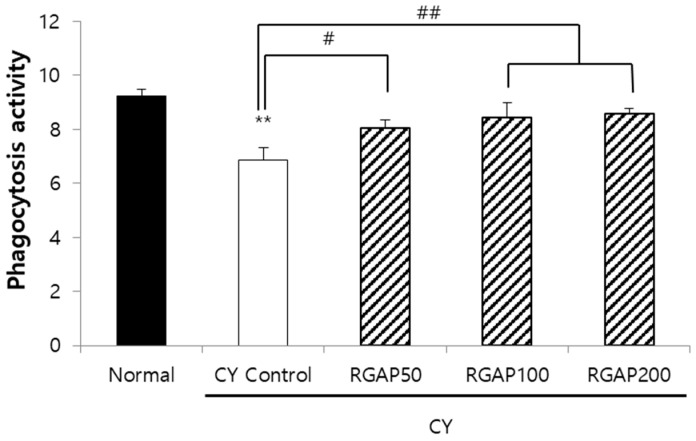
Phagocytosis activity of intraperitoneal macrophage in male Balb/c mice. Note: Mice orally administered with 50, 100, 200 mg/kg RGAP for 10 days, respectively. Normal and CY control groups were administered with distilled water. CY was administered intraperitoneally as a single dose of 50 mg/kg three days before sacrifice. Intraperitoneal macrophage was collected for test phagocytosis activity. The results are presented as the mean ± SD of eight animals per group. Values are significantly different from normal at ** *p* < 0.01. Significant differences from CY control groups presented by ^#^
*p* < 0.05 or ^##^
*p* < 0.01. CY: cyclophosphamide, RGAP: red ginseng acidic polysaccharide.

**Figure 4 molecules-25-03569-f004:**
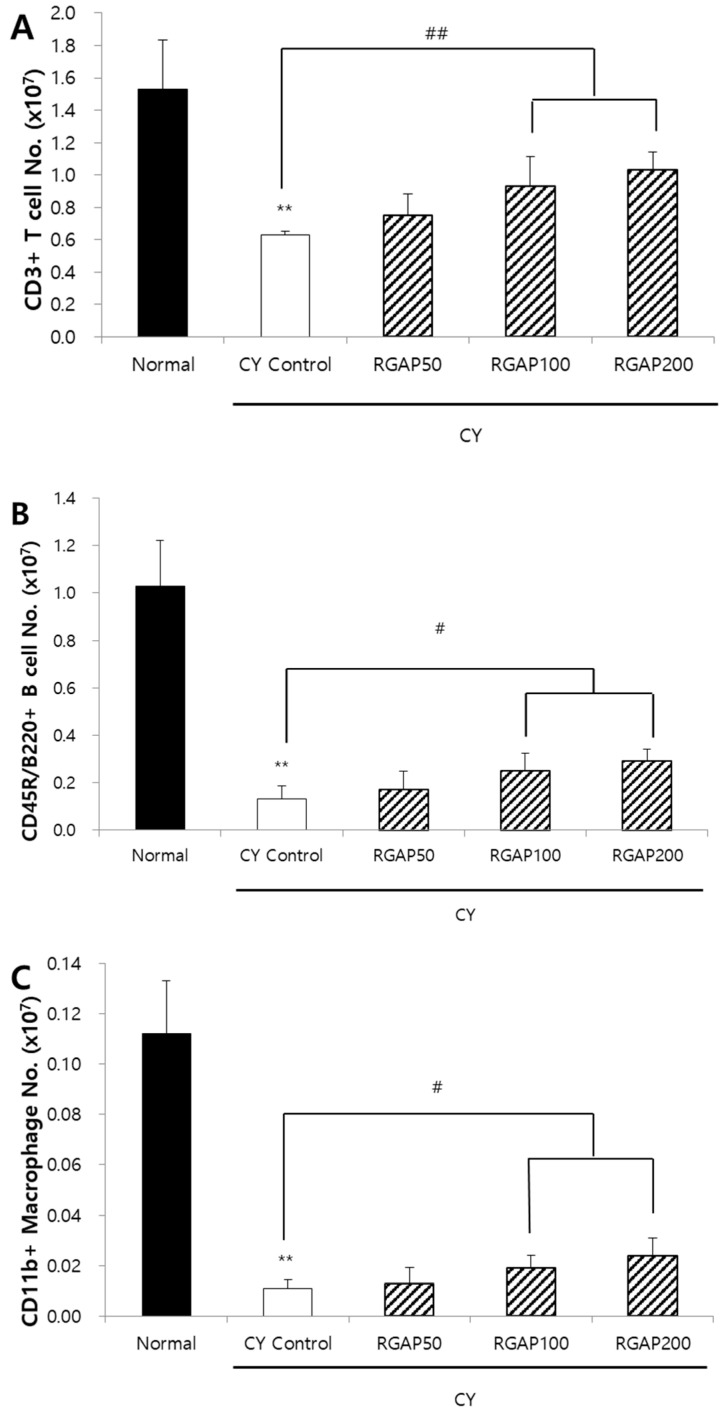
Splenic cell subtype analysis. (**A**) CD3+ T cells absolute number (10^7^ cells); (**B**) CD45R/B220+ B cells absolute number (10^7^ cells); and (**C**) CD11b+ Macrophage cells absolute numbers (10^7^ cells). Note: Male Balb/c mice orally administered with 50, 100, 200 mg/kg RGAP for 10 days, respectively. Normal and CY control groups were administered with distilled water. CY was administered intraperitoneally as a single dose of 50 mg/kg three days before sacrifice. Spleen was obtained and prepared single cell for FACS analysis. The results are presented as the mean ± SD of eight animals per group. Values are significantly different from normal at ** *p* < 0.01. Significant differences from CY control groups presented by ^#^
*p* < 0.05 or ^##^
*p* < 0.01. CY: cyclophosphamide, RGAP: red ginseng acidic polysaccharide.

**Figure 5 molecules-25-03569-f005:**
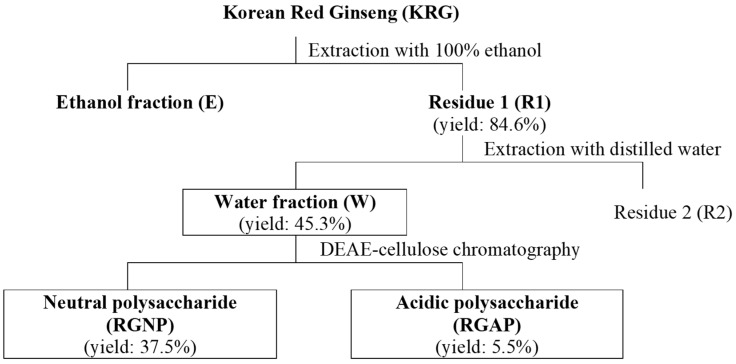
Fraction sample preparation. First fraction: ethanol fraction (E) and residue 1 (R1); second fraction: water fraction (W) and residue 2 (R2); and third fraction: red ginseng neutral polysaccharide fraction (RGNP) and red ginseng acidic polysaccharide fraction (RGAP).

**Table 1 molecules-25-03569-t001:** Chemical compositions of Korean red ginseng powder fractions (%).

Group	W ^1^	RGNP ^2^	RGAP ^3^
Glucose	2.40	1.14	N/D ^4^
Fructose	1.70	1.70	N/D
Sucrose	1.53	1.70	N/D
Maltose	5.10	5.80	N/D
Acidic polysaccharides	7.46	N/D ^5^	57.25
Ginsenosides ^5^	6.11	5.51	N/D

^1^ W: Fraction obtained by reextracting the residue obtained by ethanol extraction of six-year-old Korean red ginseng powder. ^2^ RGNP: red ginseng neutral polysaccharide. ^3^ RGAP: red ginseng acidic polysaccharide. ^4^ N/D: not detected. ^5^ Ginsenosides indicate the total percentage of Rg1, Re, Rf, Rg2(S), Rg2(R), Rb1, Rc, Rb2, Rd, Rg3(S) and Rg3(R).

**Table 2 molecules-25-03569-t002:** Organ weight in male Balb/c mice.

Experimental Group (mg/kg)	Absolute Organ Weight	Relative Organ Weight (g/100 g of Body Weight)		
	Spleen	Thymus	Spleen	Thymus
Normal	112.8 ± 0.016	55.9 ± 0.008	0.47 ± 0.0007	0.23 ± 0.0003
CY Control	84.4 ± 0.013 **	42.7 ± 0.009 **	0.37 ± 0.0005 **	0.17 ± 0.0004 **
CY + W100	88.3 ± 0.013	43.1 ± 0.007	0.37 ± 0.0007	0.18 ± 0.0003
CY + RGNP100	92.6 ± 0.015	44.4 ± 0.004	0.38 ± 0.0006	0.19 ± 0.0002
CY + RGAP100	94.3 ± 0.009	45.8 ± 0.005	0.4 ± 0.0004	0.19 ± 0.0002

Effects of orally administered 100 mg/kg of W, RGNP, and RGAP on absolute and relative organ weight. Mice orally administered with 100 mg/kg of test samples (normal: distilled water (DW); CY control: DW; CY+W100: water fraction; CY+RGNP100: red ginseng neutral polysaccharide; and CY+RGAP100: red ginseng acidic polysaccharide) for 10 days, respectively. Except for the normal group, CY was intraperitoneally injected 50 mg/kg five days before sacrifice. On the following day of CY injection, mice were immunized with 2.5 × 10^8^ SRBCs per mouse intraperitoneal injection. The results are presented as the mean ± SD of eight animals per group. Values are significantly different from normal at ** *p* < 0.01. W: water fraction, RGNP: red ginseng neutral polysaccharide, RGAP: red ginseng acidic polysaccharide, SRBC: sheep red blood cell.

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
