# Peer review of "Immune Activity of Polysaccharide Fractions Isolated from Korean Red Ginseng"

_molecules, 2020, doi:10.3390/molecules25163569_

Round 1

Reviewer 1 Report

  1. Page 6, line 239-242: The content of glucose, fructose, sucrose, and maltose in RGNP was 1.14%, 1.70%, 1.70%, and 5.80%, respectively, showing higher content compared to W, and acidic polysaccharides were not detected. Meanwhile, the content of ginsenoside was 5.51% in RGNP, which is higher compared to its content in W.

The authors should clarify how the contents were measured. Most importantly, how authors came to the conclusion that RGNP and RGAP showed a higher concentration of measured components compare to W? Figure 1 clearly demonstrates RGNP and RGAP were derived from W.

  1. Page 6, line 245-260: Authors should discuss why recovery by W is less compared to RGNP and RGAP. It is confusing as RGNP and RGAP derived from W. Moreover, the Authors should include positive control and negative control to the experiments.
  2. The authors should give details about the figure in figure legends. In the current version, it is very hard to follow figures because of the lack of details in figure legends.

Author Response to Reviewer 1

  1. Page 6, line 239-242: The content of glucose, fructose, sucrose, and maltose in RGNP was 1.14%, 1.70%, 1.70%, and 5.80%, respectively, showing higher content compared to W, and acidic polysaccharides were not detected. Meanwhile, the content of ginsenoside was 5.51% in RGNP, which is higher compared to its content in W.

The authors should clarify how the contents were measured. Most importantly, how authors came to the conclusion that RGNP and RGAP showed a higher concentration of measured components compare to W? Figure 1 clearly demonstrates RGNP and RGAP were derived from W.

Thank you for your feedback. The fractions were isolated from the Korean red ginseng powder. Because the process of manufacturing Korean red ginseng and the methods for preparing fractions were not sufficiently described in our original submission, two subclauses were added to Clause 2. Method: 2.1. Manufacture of Korean Red ginseng (line nos. 108–116) and 2.2. Production of fractions (line nos. 117–140). Concerning methods for the analysis of components found in each fraction, the analysis methods of ginsenosides and acidic polysaccharides are described in detail in Subclause 2.3: Component analysis (line nos. 143–181).

In reviewing the paper for this revision request, we found a major typographical error in Table 1 regarding the chemical compositions of Korean red ginseng powder fractions. Specifically, the decimal points for all components were misplaced. We thank the reviewers and the editor for the opportunity to address this. Rest assured that all errors in Table 1 were corrected (line nos. 182–189).

Even though not described in the draft, the yields of RGNP and RGAP obtained from W reach 37.5% and 5.5%, respectively. We present the yield of W, RGNP, and RGAP in Figure 1. The percentages (%) in Table 1 resulted from the analysis of each component of W, RGNP, and RGAP, all of which are assumed to be 100. Therefore, when the three fractions are compared with each other, relatively more of the specific components are found in RGNP and RGAP, which were separated from the DEAE-cellulose column, than in W. Following your advice, the yields of each fraction were added to aid readers’ understanding.

Korean Red Ginseng (KRG)  

Extraction with 100% ethanol  

Ethanol fraction (E)

Residue 1 (R1)

(yield: 84.6%)

Extraction with distilled water        
Water Fraction (W)
(yield: 45.3%)
Residue 2 (R2)  

DEAE-cellulose chromatography        

Neutral polysaccharide (RGNP)
(yield: 37.5%)

Acidic polysaccharide (RGAP) (yield: 5.5%)

Figure 1. Fraction samples preparation. First fraction: ethanol fraction (E) and residue 1 (R1); second fraction: water fraction (W) and residue 2 (R2); and third fraction: red ginseng neutral polysaccharide fraction (RGNP) and red ginseng acidic polysaccharide fraction (RGAP).

  1. Page 6, line 245-260: Authors should discuss why recovery by W is less compared to RGNP and RGAP. It is confusing as RGNP and RGAP derived from W. Moreover, Authors should include positive control and negative control to the experiments.

Thank you for the detailed feedback. The three samples were administered at the same dose of 100 mg/kg. The thymic and splenic weights reduced by CY were found recovering effectively. W showed the highest recovery rate, followed by RGNP and then RGAP, but it was not significant compared to CY control. Despite the insignificance, it can be inferred that this tendency is due to polysaccharides most contained in RGAP. This supposition was added to Clause 4: Discussion (line nos. 472–478).

In this study, W, RGNP, and RGAP did not significantly affect the restoration of thymic and splenic weights, which were reduced by deteriorated immunity. Note that, however, they showed a tendency to recover the weights compared to CY control. The recovery rate of the spleen and thymus was lower in W, which had a higher acidic polysaccharide content than RGNP. This was attributed to individual differences, and there was no significant difference between the two groups. Hyun et al. [32] also reported that KRG contributed to the recovery of thymic and splenic weights reduced by the weakened immune system [46].

In this experiment, groups were intraperitoneally injected with only cyclophosphamide (CY) to reduce the immunity. CY control was involved in all tests as the negative control; in addition, the CY control groups shown in the text, tables, and figures serve as negative controls.

The reason for not using positive controls that enhance the recovery and activity of the immune organs whose functions were reduced by CY is that Korean red ginseng was used as a sample. Korean red ginseng has been recognized by the Ministry of Food and Drug Safety, formerly known as the Korea Food and Drug Administration (KFDA), for its ability to help “improve immunity.” In addition, given that many studies have revealed the immune function of RG and the fractions used in this study are derived from RG, no other materials were used for positive controls. We described Korean red ginseng’s immunity ability in the Introduction (line nos. 38–42).

KRG has been recognized by the Ministry of Food and Drug Safety, formerly known as the Korea Food and Drug Administration (KFDA), as a health functional food with six health benefits: boosting immunity, overcoming fatigue, improving memory, improving blood circulation, alleviating women's menopausal symptoms, and promoting antioxidant activity [4].

  1. Authors should give details about figure in figure legends. In the current version, it is very hard to follow figures because of the lack of details in figure legends.

Thank you for your suggestion. To increase the legibility of all figures, CY administration is expressed in all figures, and the statistical marks were also modified for better understanding. In addition, the figure captions were corrected for easier understanding. To be specific, when there are three figures, a more detailed description of each figure is given. In line with this, we revised all figure captions.

Reviewer 2 Report

Submitted manuscript covers interesting topics and presents valuable data worth publishing. However, prior the publication the manuscript needs to undergo essential changes. The introduction is sufficient, however, little information regarding the health benefits of red ginseng was provided. Initial working material and way of acquisition (red ginseng) should be described. Methods were described accurately. Results were presented clearly. Discussion should be subtracted from the conclusions and a brief summary of the article should be provided. Literature is quite comprehensive, however, most of the provided references (36/47) are more than ten years old and only two were published within recent five years. The authors should state their conflict of interest !!!

Detailed comments:

Line 72-74 literature should be provided;

Line 110 please state RCF / G instead of rpm;

Line 150 repetition of “to” in lux range;

Line 157 I’m suggesting starting new sentence form next line;

Line 162 same suggestion as above;

Line 177 please state RCF / G instead of rpm;

Line 185 same suggestion as above;

Line 215 same suggestion as above;

Lines 384-439 Please separate discussion and conclusions, this part is rather discussion, please provide general summary of the results in conclusions.

Author Response to Reviewer 2

  1. Submitted manuscript covers interesting topics and presents valuable data worth publishing. However, prior the publication the manuscript needs to undergo essential changes. The introduction is sufficient, however, little information regarding the health benefits of red ginseng was provided. Initial working material and way of acquisition (red ginseng) should be described

Thank you for the detailed feedback. Korean red ginseng has been recognized by the Ministry of Food and Drug Safety, formerly known as the Korea Food and Drug Administration (KFDA), as a health functional food with six health benefits. Description regarding these six health effects was added (line nos. 38–42).

KRG has been recognized by the Ministry of Food and Drug Safety, formerly known as the Korea Food and Drug Administration (KFDA), as a health functional food with six health benefits: boosting immunity, overcoming fatigue, improving memory, improving blood circulation, alleviating women's menopausal symptoms, and promoting antioxidant activity [4].

In addition, the process of manufacturing Korean red ginseng in conjunction with methods for fraction preparation was additionally described to help readers understand the initial stage of preparing materials used for research. Subclause 2.1 (line nos. 108–116) was added under the Methods section to describe the process of manufacturing Korean red ginseng. A paragraph was also inserted before the paragraph regarding W to explain the production of the fractions (line nos. 118–127), as follows.

2.1. Manufacture of Korean red ginseng

In this experiment, Panax ginseng was grown in Gyeonggi-do, Icheon (South Korea), for six years. The manufacturing process of Korean red ginseng was certified in April 2017 by the International Organization for Standardization (ISO) and officially approved globally (ISO 19610); the process is as follows: sorting six-year-old fresh ginseng roots based on the thickness of the main root before washing; steaming at a temperature of 90℃ to 100℃ for at least 80 to 100 min for starch gelatinization; and, lastly, drying with hot air at 45℃ to 55℃ and in the sun until the moisture content is 15.5% or less. KRG powder is a product of pulverizing dried KRG (120 mesh or less) with a moisture content of less than 3% to 6%.

2.2. Production of fractions

To manufacture fractions, 1 kg of KRG powder and 3 L of 100% ethanol (EtOH) were mixed and extracted using ultrasonic extraction 2 hr at 40℃. The supernatant was collected after extraction. Precipitated residue was added 2 L of 100% EtOH and extracted ultrasonic waves 2 hr at 40℃ again. The above method was performed three times. Upper solutions were collected, vacuum evaporated 50℃, and lyophilized three days to make the ethanol fraction (E). Resides were collected and dried with an air dryer at 45℃ for 12 hr, from which Residue 1 (R1) was obtained. R1 was dissolved with DW and extracted for 4 hr using ultrasonic waves at room temperature. Upper solutions were obtained after centrifuge, and the same procedure was performed two more times. Afterward, upper solutions were collected, vacuum evaporated in a 60℃ water bath, and lyophilized to obtain the water fraction (W).

  1. Methods were described accurately. Results were presented clearly. Discussion should be subtracted from the conclusions and a brief summary of the article should be provided

Thank you for the recommendation. Following this, the Conclusion and Discussion clauses were divided into two sections. In addition, we supplemented our discussion for clarity (line nos. 485–496). A summary of this paper was also provided in the Conclusion clause (line nos. 509–515).

  1. Literature is quite comprehensive, however, most of the provided references (36/47) are more than ten years old and only two were published within recent five years. The authors should state their conflict of interest !!!

Thank you for your comments. Given the very limited number of studies on RG-derived acidic polysaccharides conducted, the references attached hereto are not recent versions. However, to address this issue, we added the following recent versions of research on acid polysaccharides and red ginseng. Finally, we made sure that 25 out of the 47 references are published within 10 years, and 10 out of all references are published within 5 years. 

Moreover, when this paper was first submitted, this statement was included: “Conflicts of interest: The authors declare no conflict of interest.” However, to ensure no confusion, we made sure that this was included in this submission.

Detailed comments:
Line 72-74 literature should be provided;

The relevant literature has been added (line nos. 92–95).

However, although various studies on the immune activity of KRG have been done, the components that contribute to the enhancement of immunity have not been identified. It can be inferred from existing literature that RGAPs carry out immune-physiological activity, but no studies have systematically proven such a hypothesis [3].

Line 110 please state RCF / G instead of rpm;

Thank you for the comment. We changed “rpm” to “rcf” throughout the paper.

Line 150 repetition of “to” in lux range;

We changed “illuminance adjusted to 150 to 300 Lux” to “illuminance was between 150 and 300 Lux” (line no. 209).

Line 157 I’m suggesting starting new sentence form next line;

We made this into a new paragraph.

Line 162 same suggestion as above;

We made this correction.

Line 177 please state RCF / G instead of rpm;

Thank you for the comment. We changed instances of “rpm” to “rcf” throughout the paper.

Line 185 same suggestion as above;

We made the same correction as above (line no. 244).

Line 215 same suggestion as above;

We made the same correction as above (line no. 274).

Lines 384-439 Please separate discussion and conclusions, this part is rather discussion, please provide general summary of the results in conclusions.

Thank you for the recommendation. The Discussion and Conclusion clauses have been separated to supplement the logical structure of this paper (see new Discussion clause in line nos. 449–508).

KRG is an exemplary herbal medicine known for its pharmacological effects in Asia, especially Korea and China [37], as many studies have shown its role as herbal medicine in controlling anticancer, cardiovascular disease, immunity, diabetes, and skin activity [3841]. In particular, its most representative pharmacological effect is the regulation of the immune system and protection from external infections (viruses, cells, diseases) [29]. Active ingredients of KRG include ginsenosides, flavonoids, polyphenols, and polysaccharides, which are generally divided into RGNP and RGAP [42]. The acidic polysaccharide refers to a polysaccharide with a molecular weight of 10,000 to 15,000 containing a large amount of acidic sugar, like galacturonic acid, glucuronic acid, and mannuronic acid. Such RGAPs are also known to have a greater effect on the immune system than RGNPs [26].

A previous study has found that the acidic polysaccharides isolated from Panax ginseng promote the proliferation of T cells and B cells and affect the immune system [22]. In addition, cytotoxic activity against B16 melanoma cells was induced when macrophages were treated with polysaccharides, and the secretion of tumor necrosis factor-α (TNF-α), interleukin-1β (IL-1β), IL-6, and interferon-γ (IFN-γ) was promoted [43]. Moreover, many studies confirmed that RGAPs have various immune activity effects [44, 45]. Although there are many studies on the immune activity of acidic polysaccharides, these studies consider the crude form of the polysaccharide. In addition, no systematic study has been done on the preparation of fractions and comparison of the activity of each fraction to identify components that show immune activity in a single root of KRG, as in this study.

This study is meaningful in that it systematically identified the components contributing to the immune activity of KRG. Fractions were prepared from KRG powder for each step, and the immunological activities of the fractions were compared to select a superior fraction.

In this study, W, RGNP, and RGAP did not significantly affect the restoration of thymic and splenic weights, which were reduced by deteriorated immunity. Note that, however, they showed a tendency to recover the weights compared to CY control. The recovery rate of the spleen and thymus was lower in W, which had a higher acidic polysaccharide content than RGNP. This was attributed to individual differences, and there was no significant difference between the two groups. Hyun et al. [32] also reported that KRG contributed to the recovery of thymic and splenic weights reduced by the weakened immune system [46]. RGAP turned out to have the highest AFC formation, followed by W and then RGNP. This meant that a fraction with more acidic polysaccharides is more capable of producing antibodies. As shown in this study, Park et al. [28] also demonstrated that acidic polysaccharides isolated from KRG promoted antibody production in mice. As a result, it was found that the higher content of acidic polysaccharides showed relatively higher AFCs when compared with each fraction’s components. This finding could lead to the reasoning that among different KRG components, RGAPs are polysaccharides that contribute to immunoreactivity.

This study reviewed further work regarding RGAP’s immune activities such as AFCs, phagocytosis activity, and splenocyte immune cell subtype analysis. Here, RGAP groups showed significantly increased AFCs in all groups dose-dependently than immunosuppressed CY control. Lee et al. [47] studied KRG polysaccharides’ immune effect, and their results showed increased AFCs compared to negative control. The present study shows that phagocytosis activity significantly increased compared to CY control in all doses of RGAP groups. Phagocytosis is a crucial physiological process that is characterized by the ingestion of foreign particles and the killing of bacteria by phagocytic leukocytes, including macrophages; it serves as the first line of host defense against pathogens. This study’s results indicated that RGAP potentially enhanced phagocytosis to defend the host from infection. Moreover, the splenic absolute number of CD3+ T cells, CD45R/B220+ B cells, and CD11b+ macrophage cells were significantly increased at RGAP 100, 200 mg/kg groups. These results indicated that RGAP exhibited immune-enhancing effects. Byeon et al. [37] investigated the mechanism by which RGAP stimulates the immune response. RGAP treatment on macrophages showed that it activates transcription factors, such as nuclear factor kappa-light-chain-enhancer of activated B cells (NF-κB), and activator protein 1 (AP-1) and upstream signaling enzymes, such as extracellular signal-related kinase (ERK), to activate macrophages and induce immune activity. Similarly, in this study, RGAP was found to activate immune cells (T cells, B cells, and macrophages) and increase their number to induce immune activity.

Clear composition and structural activity of acidic polysaccharides have not been identified yet. Also, acidic polysaccharides' immune active mechanisms were still insufficient. Therefore, further study needs to be done on the components of polysaccharides derived from KRG. The RGAP found to have excellent immune activity in this study showed the possibility of development into a functional food and pharmaceutical supplement, replacing the conventional immunomodulatory substances that present toxicity and side effects.

In addition, a summary of this study has been prepared in the Conclusion clause (line nos. 510–515)

KRG was fractionated with ethanol and water, the component content of each fraction was analyzed, and the immune activity of each fraction in the animal was compared. As a result, the more acidic the polysaccharide content was, the higher the immune activity was. It could be assumed that the immunoreactive component of KRG is an acidic polysaccharide. In addition, this result is thought to support the results of many papers that have suggested that the immunoreactive component of KRG is an acidic polysaccharide.

Round 2

Reviewer 1 Report

Authors have improved the manuscript in such a way that it is suitable for publication.

Reviewer 2 Report

The manuscript has been substantially improved and can be recommended for publication in the present form. However, prior to the publication I insist that authors declare their evident conflict of interest. Authors affiliate with Korea Ginseng Corporation that is a producer of Red Ginseng, therefore for clarity and transparency this should be mentioned.